



# Projecting the evolution of the Northern Patagonian Icefield until the year 2200

Marius Schaefer[1,2], Ilaria Tabone[1,3], Ralf Greve[4,5], Johannes J. Fürst[1], and Matthias Braun[1]

[1]Institute of Geography, Friedrich-Alexander Universität Erlangen-Nürnberg, Erlangen, Germany
[2]Institute of Physical and Mathematical Sciences, Austral University of Chile, Valdivia, Chile
[3]Department of Geophysics, University of Concepción, Chile
[4]Institute of Low Temperature Science, Hokkaido University, Sapporo, Japan
[5]Arctic Research Center, Hokkaido University, Sapporo, Japan

**Correspondence:** Marius Schaefer (mschaefer@uach.cl)

**Abstract.** The Northern Patagonian Icefield (NPI), Chile, is the second-largest ice mass in the Southern Hemisphere outside Antarctica and a major remnant of the Patagonian ice sheet from the Last Glacial Period. It is located in the Southern Andes, which is among the world's glacierized regions with the most negative specific mass balances. The NPI is a highly dynamic system, with high amounts of accumulation and ablation, and includes Glaciar San Rafael, the tidewater calving glacier closest to the equator.

Using the ice-sheet model SICOPOLIS, we reproduce the dynamical state and observed changes of the NPI in the early 21st century and project its evolution until 2200. Calving is represented by prescribing an additional mass loss for ocean-terminating grid cells (Glaciar San Rafael). A spin-up experiment generates an icefield comparable to conditions around the year 2000, which we then force with present-day and projected surface mass balance under climate scenarios SSP1-2.6 and SSP5-8.5.

In the committed mass loss run, the NPI stabilizes by 2100 at around 75% of its current volume. Under climate change scenarios, mass loss accelerates from the mid-21st century and continues until 2200, despite assuming constant climate during the final century. The NPI exhibits a response time of approximately 100 years, highlighting the need for caution when interpreting current trends. By 2200, the remaining volume strongly depends on the emission pathway: $64 \pm 10\%$ under SSP1-2.6 versus $32 \pm 14\%$ under SSP5-8.5. These results confirm that for Patagonia, as found elsewhere, every fraction of a degree of warming matters

## 1 Introduction

Glaciers are shrinking all over the world and, under business-as-usual emission scenarios, many mountain ranges are projected to be largely deglacierized by the end of the 21st century (Marzeion et al., 2020; Rounce et al., 2023). In the Southern Andes, the specific mass balance of glaciers was estimated to be among the most negative worldwide (Zemp et al., 2019). Due to the large amount of ice in the Southern Andes, which is mostly stored in a few large icefields, the projected relative mass loss until the end of the century is moderate: the Glacier Model Intercomparison Experiment (GlacierMIP) (Marzeion et al., 2020) suggests losses between 21% under RCP2.6 and 41% under RCP8.5 with respect to the estimated glacier mass in 2015.



The global models that participated in GlacierMIP have reduced capacities to reproduce the complex dynamics of ice bodies like the Northern Patagonian Icefield (NPI) and the Southern Patagonian Icefield (SPI): when projecting the geometry change of the ice body, they rely on volume-area scaling or flowline models (Marzeion et al., 2020). Only two of the eleven models parameterize frontal ablation, which can make a very important contribution to the total ablation of the large calving glaciers of the Patagonian Andes (up to 91%, Minowa et al. (2021)).

To our knowledge, only for two glaciers of the Southern Andes projections have been carried out using higher-order flow models: Glaciar San Rafael, the largest glacier of the NPI (Collao-Barrios et al., 2018), and the Mocho-Choshuenco ice cap in the Chilean Lake District (Scheiter et al., 2021). Collao-Barrios et al. (2018) used the finite-element ice-flow model Elmer/Ice to model the behaviour of San Rafael Glacier. Assuming a fixed glacier outline and neglecting elevation feedbacks for the surface mass balance (SMB), they project a committed mass loss of 33.7 Gt, which corresponds to 14% of the estimated glacier mass of the year 2000. Scheiter et al. (2021) modelled the dynamics of the Mocho-Choshuenco ice cap using the SImulation COde for POLythermal Ice Sheets (SICOPOLIS). Temperature projections from 23 models of Phase 5 of the Coupled Model Intercomparison Project (CMIP5) (Taylor et al., 2012) and a temperature-dependent parameterization of the equilibrium line altitude (ELA) were used to project the future SMB of the ice cap. They projected a relative volume loss between 56% (RCP2.6) and 97% (RCP8.5) at the end of the 21st century. The actual and future SMB of the entire NPI was studied by Schaefer et al. (2013) and Bravo et al. (2021). Schaefer et al. (2013) used an SMB model that considers solid precipitation as accumulation and calculates surface ablation as a function of incoming solar radiation, albedo and temperature. The model is driven by statistically downscaled global climate datasets: NCEP/NCAR for the period 1975–2011 and ECHAM-5 A1B scenario for the future. They found a slightly negative SMB of the NPI with an increasing trend during the recent past (1975–2011). A decreasing trend of the SMB was projected for the second half of the 21st century due to an increase in surface ablation and a decrease in accumulation. The modelled SMB showed a significant correlation with the projected annual mean temperature over the model domain (see Sect. 3.2.2). Bravo et al. (2021) used dynamically downscaled outputs of the MPI-ESM-MR Earth system model obtained with the regional climate model RegCM4.6 at 10-km spatial resolution between 1976 and 2005 and for the RCP2.6 and RCP8.5 scenarios, between 2005 and 2050. These products were used directly to estimate snow accumulation and subsequently to feed an energy balance model to estimate the ablation and its changes over both Patagonian icefields. For the NPI they projected an insignificant decrease in the SMB during the first half of the 21st century ($-0.01$ m water equiv. $\mathrm{a}^{-1}$) under the RCP2.6 scenario and a slightly higher and significant decrease under the RCP8.5 scenario ($-0.08$ m water equiv. $\mathrm{a}^{-1}$).

Here we study the complex ice dynamics of the NPI using the SImulation COde for POLythermal Ice Sheets (SICOPOLIS) (SICOPOLIS Authors, 2025) to better understand its current changes and project its evolution until the year 2200. We realize a spin-up to generate an icefield similar to the state of the NPI in the year 2000. We calibrate key model parameters by comparing our simulations to remotely sensed observations of elevation change (2000–2014) (Braun et al., 2019) and surface velocity (Mouginot and Rignot, 2015; Friedl et al., 2021). We chose the best set of model parameters and continue the simulations into the future with a constant (current) SMB to obtain the committed mass loss. In a next step, we use the temperature dependence of the projected SMB by Schaefer et al. (2013) to generate SMB anomalies for the CMIP6 model mean temperature projection





(mean of NPI grid cells) in the 21st century for the pathways SSP1-2.6 and SSP5-8.5 (corresponding to RCP2.6 and RCP8.5 in CMIP5). For the 22nd century, we generate SMB anomalies for the NPI without trend by arbitrarily choosing one year of

60 the last decade of the 21st century using the corresponding (model mean) temperature projection. Using these SMB anomalies, we project the evolution of the NPI until 2200 under the different scenarios of climate change.

## 2   Study site

The NPI with an area of $3700 \, \text{km}^2$ (Meier et al., 2018) is the second largest ice mass in the Southern Hemisphere outside of Antarctica. Similar to the European Alps, it is located at moderate latitudes (Figure 1), which are considerably lower than in

65 other glaciated regions with maritime influence (Alaska or Northern Europe).

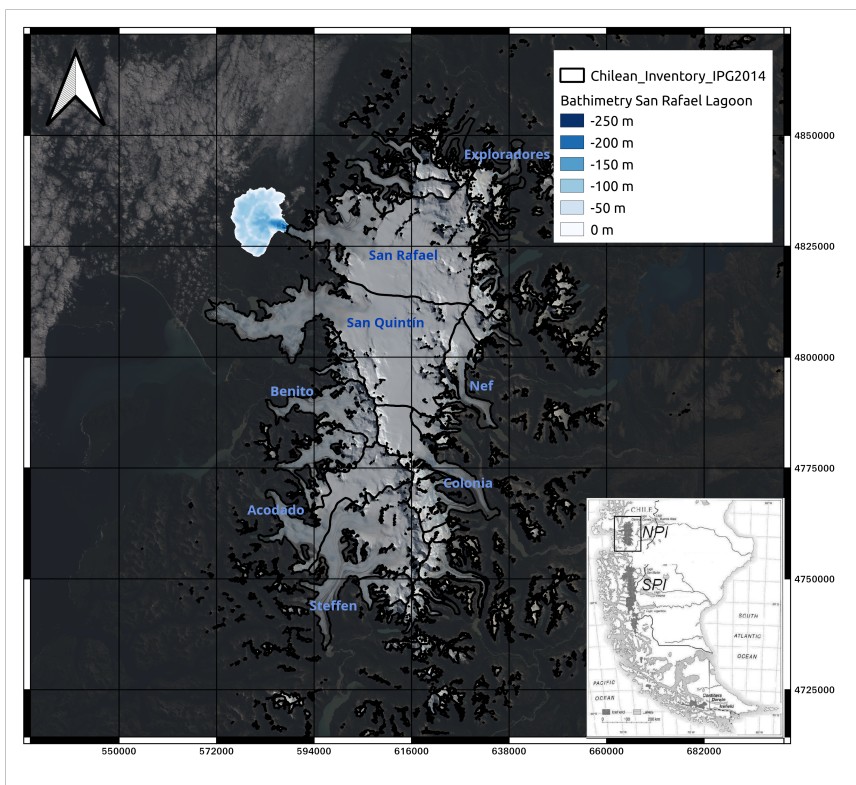

**Figure 1.** Main figure: NPI and some of its most important glacier catchments and bathymetry of San Rafael Lagoon; glacier outlines from Barcaza et al. (2017) that are consistent with Randolph Glacier Inventories version 6.0 & 7.0 for the main catchments of the NPI; bathymetry data from Koppes et al. (2011); coordinates are UTM 18S, superposed grid cells are $20 \times 20 \, \text{km}$; background satellite image is a true-color composite of Landsat image acquired on 12 March 2016;.Inset: location of the NPI in southern South America.





Glaciar San Rafael (the only tidewater calving glacier of the NPI) is the tidewater calving glacier closest to the equator. For the period 2000–2019 Minowa et al. (2021) found a mean annual frontal ablation of $0.99 \pm 0.29 \, \mathrm{Gt \, a^{-1}}$ for Glaciar San Rafael, which corresponds to 36% of its total ablation. Fürst et al. (2024) found a slightly higher frontal ablation value for San Rafael Glacier of $1.26 \pm 0.21 \, \mathrm{Gt \, a^{-1}}$ for the beginning of the 21st century (using glacier geometries of the year 2000 and ice velocities from Mouginot and Rignot (2015)). Several other glaciers of the NPI calve into rather small pro-glacial lakes, and their frontal ablation accounts for less than 20% of their total ablation (Minowa et al., 2021). Measurements were performed over the outlet tongues of the glaciers Colonia and Nef using airborne radar systems, and ice thicknesses of over $700 \, \mathrm{m}$ were detected (Blindow et al., 2012a). On the plateau area of the NPI, ice thickness was inferred from gravimetry measurements reaching values of up to $1400 \, \mathrm{m}$ (Millan et al., 2019). Recently, the ice thickness of the NPI was modelled using a mass conservation approach, which considered all available ice thickness measurements over the NPI (Fürst et al., 2024).

## 3 Methods

### 3.1 Ice-flow model

The three-dimensional, dynamic and thermodynamic model SICOPOLIS was originally created in a version for the Greenland ice sheet (Greve, 1997a, b). Since then, the model has been developed continuously and applied to problems of past, present and future glaciation of Greenland, Antarctica, the entire Northern Hemisphere, the polar ice caps of the planet Mars, and other places, resulting in more than 150 publications in the peer-reviewed literature (https://www.sicopolis.net, last access: 5 June 2025).

Mainly developed for ice sheets, smaller ice bodies have also been modelled with SICOPOLIS, namely the Austfonna ice cap (Dunse et al., 2011) and the Mocho-Choshuenco ice cap (Scheiter et al., 2021). For this study, we apply SICOPOLIS v25 (SICOPOLIS Authors, 2025) to the NPI, using a polar stereographic projection with the WGS84 reference ellipsoid, standard parallel $47°\mathrm{S}$ and central meridian $75°\mathrm{W}$ (same as for UTM 18S). The stereographic plane is spanned by the Cartesian coordinates $x$ (easting) and $y$ (northing), discretized by a regular (structured) grid with resolution $\Delta x = 0.9 \, \mathrm{km}$. In the vertical, we use a terrain-following coordinate $\zeta$ (equal to 1 at the ice surface and 0 at the ice base; "sigma transformation") instead of the physical coordinate $z$, with 81 layers in the ice domain, concentrating towards the base.

For the ice rheology, we use the regularized Glen flow law in the form of Greve and Blatter (2009, Sect. 4.3.2). The dynamics of grounded ice is modelled by the hybrid shallow-ice–shelfy-stream formulation. In this mode, the 3D velocity field is computed by the Shallow Ice Approximation if the local slip ratio (basal to surface velocity) is less than a threshold value, chosen as 50%. Otherwise, the velocity field is computed as a weighted average between the Shallow Ice Approximation and the shelfy-stream approximation (Bernales et al., 2017; Greve et al., 2020). Ice thermodynamics is modelled by the one-layer melting-CTS enthalpy scheme (CTS: cold-temperate transition surface; Blatter and Greve, 2015; Greve and Blatter, 2016). The temperature-dependent rate factor for cold ice is computed following Cuffey and Paterson (2010) (Sect. 3.4.6), and the water-content-dependent rate factor for temperate ice following (Lliboutry and Duval, 1985).





The ice surface is assumed to be traction-free. Basal sliding at the glacier bed is parameterized by a linear sliding law, in which the slip velocity $v_b$ is proportional to the basal shear stress $\tau_b$:

$$v_b = -C_b \tau_b. \tag{1}$$

The sliding function $C_b$ is assumed to depend on the basal temperature relative to pressure melting $T_b$ and the thickness of the basal water layer $H_w$:

$$C_b = C_b^0 \exp\left(\frac{T_b}{\gamma}\right)\left[1 + C_w(1 - \exp\left(-\frac{H_w}{H_w^0}\right))\right], \tag{2}$$

where $H_w$ is computed by a steady-state routing scheme for subglacial water that receives its input from the basal melting rate under grounded ice (Le Brocq et al., 2006, 2009), $C_b^0$ is the sliding coefficient, $C_w$ the coefficient for water-layer-enhanced sliding, $\gamma$ the sub-melt-sliding parameter and $H_w^0$ the threshold water-layer thickness. We use $C_w = 99$, $\gamma = 10\,\text{K}$ and $H_w^0 = 1\,\text{cm}$. In Figure 2 we show how the sliding function $C_b$ varies with the water-layer thickness $H_w$ at several basal temperatures using $C_b^0 = 2 \times 10^{-4}\,\text{m a}^{-1}\,\text{Pa}^{-1}$).

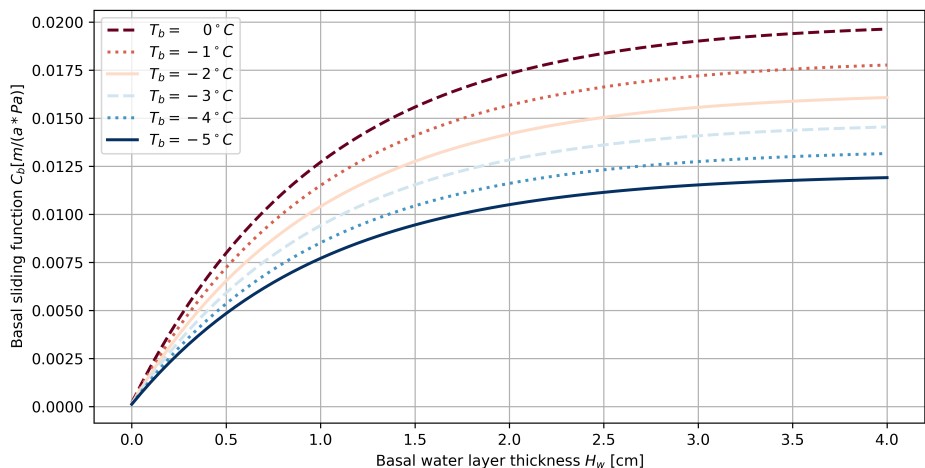

**Figure 2.** Dependence of the basal sliding function $C_b$ on the basal water-layer thickness $H_w$ at different basal temperatures $T_b'$.

Due to the rather small size of the icefield and short simulations periods, we ignore glacial isostatic adjustment and assume a rigid bed instead. In addition, we ignore the thermal inertia of the lithosphere and apply the geothermal heat flux, chosen as $q_{\text{geo}} = 65\,\text{mW m}^{-2}$, directly at the base of the ice cap. Further model parameters are listed in Table 1.

We implemented a new calving condition, which adds an additional mass loss at grid cells in contact with the ocean. In our case, this occurs only at the front of the San Rafael Glacier and we chose a constant additional mass loss per grid cell in a way that the overall modelled losses by calving fits the current observed calving flux for this glacier (Minowa et al., 2021). Since this approach turned out to generate satisfactory results for the front position of San Rafael Glacier, we did not investigate



| Quantity | Value |
|---|---|
| Density of ice, $\rho$ | $910\,\mathrm{kg\,m^{-3}}$ |
| Gravitational acceleration, $g$ | $9.81\,\mathrm{m\,s^{-2}}$ |
| Length of year, 1 a | $31\,556\,925.445\,\mathrm{s}$ |
| Power-law exponent, $n$ | 3 |
| Residual stress, $\sigma_0$ | $10\,\mathrm{kPa}$ |
| Melting temperature at low pressure, $T_0$ | $273.16\,\mathrm{K}$ |
| Clausius-Clapeyron gradient, $\beta$ | $8.7 \times 10^{-4}\,\mathrm{K\,m^{-1}}$ |
| Universal gas constant, $R$ | $8.314\,\mathrm{J\,mol^{-1}K^{-1}}$ |
| Heat conductivity of ice, $\kappa$ | $9.828\,\mathrm{e}^{-0.0057\,T[\mathrm{K}]}\,\mathrm{W\,m^{-1}K^{-1}}$ |
| Specific heat of ice, $c$ | $(146.3 + 7.253\,T[\mathrm{K}])\,\mathrm{J\,kg^{-1}K^{-1}}$ |
| Latent heat of ice, $L$ | $3.35 \times 10^5\,\mathrm{J\,kg^{-1}}$ |

**Table 1.** Physical parameters used for the simulations of this study.

more sophisticated "calving laws". Since the amount of frontal ablation at lake-terminating glaciers is small in comparison with other mass-loss terms (22% of the total frontal ablation (Fürst et al., 2024)), which accounts for around 18% of the total ablation Minowa et al. (2021)), and future changes are very difficult to predict, we decided not to focus on lake-terminating glaciers, and no calving laws were applied for these glaciers.

## 3.2 Data

### 3.2.1 Bed topography

Consistent bed topography data is one of the most crucial inputs for ice-flow modelling. For our study, the basal topography map for the ice-covered parts of the NPI was generated using a mass-conservation approach (Fürst et al., 2024). The approach assimilated available thickness measurements and relied on glacier geometry (RGI Consortium, 2017; Farr and Kobrick, 2000),

1975–2011 climatic mass balance (Schaefer et al., 2013) and 2004 velocity information (Mouginot and Rignot, 2015). For NPI thickness observations, two regional airborne gravimetry campaigns from 2012 and 2016 (Gourlet et al., 2016; Millan et al., 2019) were considered, as well as a terrestrial from 1985 (Casassa, 1987). Radar measurements of different spatial resolution were used on San Rafael, San Quintin, Soler, Nef and Colonia glaciers (Rivera, 2012; Blindow et al., 2012b; Pętlicki et al., 2023). For Laguna San Rafael, bathymetry information was available from a 2006 sonic survey (Koppes et al., 2010). According

to the generated bed topography map 78% of the NPI frontal ablation is channelized through San Rafael Glacier (Fürst et al., 2024), which justifies our approach to focus on this glacier when implementing the frontal ablation scheme in SICOPOLIS (see above).





### 3.2.2 Present and future surface mass balance

The surface mass balance (SMB) used to drive our simulations were derived from an extensive study of the interaction of
the NPI with its local climate (Schaefer et al., 2013). In this study, a regional climate model was run on a 5 km grid forced
by reanalysis data. The climate data were further downscaled statistically to feed an SMB model of intermediate complexity,
which was applied to the NPI. Model parameters were adjusted to reproduce observed mass changes from geodetic studies
and direct observation such as ablation stakes and shallow firn cores. In order to realize projections, the physical downscaling
of the reanalysis data was statistically analyzed and a statistical downscaling was applied to the projections of the ECHAM5
model (A1B) scenario (Schaefer et al., 2013). The resulting annual mean SMB of the NPI shows a strong correlation with
the projected temperature anomaly (Figure 3, left panel). In this contribution, we make use of this linear dependency and
construct projected SMB anomalies under current climate change scenarios using projected temperature anomalies in the 21st
century (Figure 3, right panel). For the projected temperature anomaly times series, we use the mean value of 33 models
of Phase 6 of the Coupled Model Intercomparison Project (CMIP6) (O'Neill et al., 2016), which are listed in detail in the
Appendix (Table A1). In order to quantify the uncertainty of the temperature projections, we calculated the standard deviations
between the temperature anomalies projected by the climate models and generated SMB anomalies corresponding to model
mean temperature anomaly projections plus/minus one standard deviation (discontinuous lines, Figure 3 right panel). It is
important to note that these standard deviations are calculated on the temperature anomalies that were calculated with respect
to the reference period 2000–2020 and not on the absolute temperatures. To be able to observe the adjustment of the NPI to the
strongly transient climate in the 21st century, for the 22nd century, we used temperature anomalies without a trend by randomly
picking a year of the last ten years of the 21st century and using the associated temperature and SMB anomalies.

### 3.2.3 Ice speed

Ice speed observations of the NPI were taken from remote sensing studies by Mouginot and Rignot (2015) for the year 2004,
and by Friedl et al. (2021) for each of the years 2014–2020. They were used to calibrate the sliding coefficient $C_b^0$ (Eq. (2))
and the enhancement factor $E$ of the flow law. The general pattern of the ice motion did not vary too much over our calibration
period 2000–2020.

### 3.2.4 Elevation change

Observed elevation changes for the early 21st century are available from geodetic surveys (e.g. (Braun et al., 2019)). The
capacity of our model to reproduce these observed elevation changes is a good indicator for the reliability of our projections
for the rest of the 21st and 22nd century. However, currently observed changes might be caused by changes in climate and
ice dynamics in the past. Additionally, it is impossible to exactly generate the state of the NPI in 2000 during the spin-up
(see below). Therefore, we do not expect to be able to reproduce exactly the observed elevation changes. We use the observed
elevation changes generated by Braun et al. (2019) for 2000–2014 to optimize model performance during the calibration period
(2000–2020).



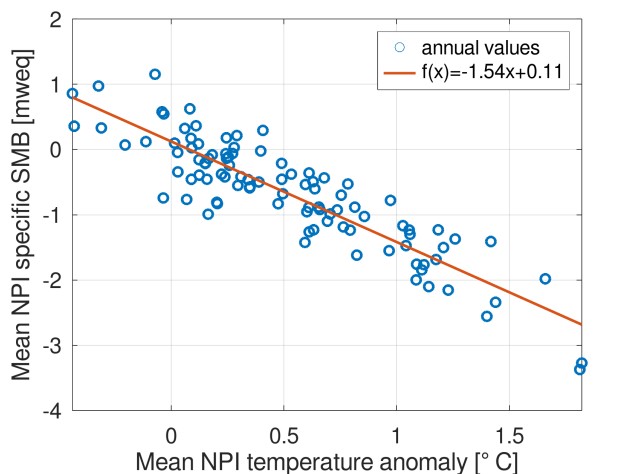 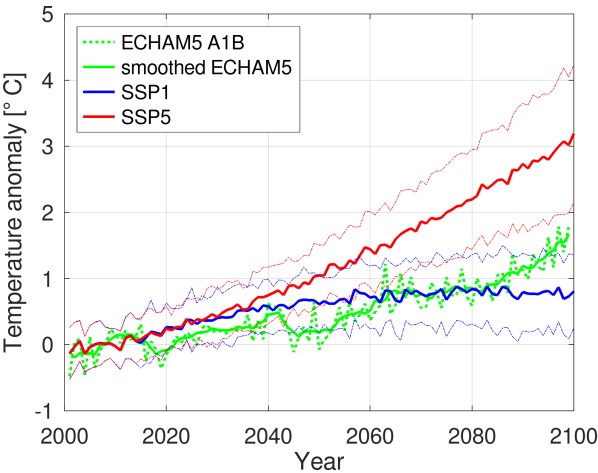

**Figure 3.** Left: Relationship of NPI averaged surface mass balance and temperature anomalies for the 21st century (from Schaefer et al. (2013)). Right: Temperature anomalies over the NPI as projected by the ECHAM5-Model (A1B scenario, green lines) (Roeckner et al., 2003) and the mean (red and blue solid lines) ± one standard deviation (discontinuous lines) of 33 CMIP6 models (Table A1). The selected reference period for the anomalies is 2000–2015.

## 3.3 Modelling workflow

In order to generate an icefield with similar characteristics as the NPI around 2000, we realized spin-up runs with a constant SMB. When using the 1975–2011 mean SMB generated by Schaefer et al. (2013), the resulting icefield has a much lower volume compared to the observed one (not shown). This indicates that the NPI in 2000 probably was not in equilibrium with the current climate. We therefore realize the spin-up with a more positive constant SMB, which, according to the linear relationship between temperature and SMB indicated in Figure 3, corresponds to a colder climate. When using a constant SMB that corresponds to a one degree colder climate, after approximately 500 years a steady state is reached that has a similar overall ice volume (maximum 3% of difference, Table 2), surface elevations (RMSD $\sim 88$ m) and surface velocities (RMSD $\sim 351$ m a$^{-1}$) (see Figures 4, 5).

In the next step, the icefield generated in the spin-up is forced with temperature anomalies for the years 2000–2020 (we chose to take them from SSP1-2.6, which is identical to SSP5-8.5 in the first 15 years) and the evolution of the icefield during this period is compared to observations of elevation change (Braun et al., 2019) and ice motion (Mouginot and Rignot, 2015; Friedl et al., 2021). The empirical model parameters ($C_b^0$, enhancement factor $E$) were varied, and a comparison of the simulation results with observations was realized (see Table 2). Using the best empirical model parameter set (defined below), the simulations were driven by the different 2000–2100 climate change scenarios, which were continued until 2200 using temperature anomalies from 2090–2099 (see above) for the 22nd century. In order to obtain an estimate for the committed mass loss, a simulation was run with a constant SMB (1975–2011 mean, Schaefer et al. (2013)).



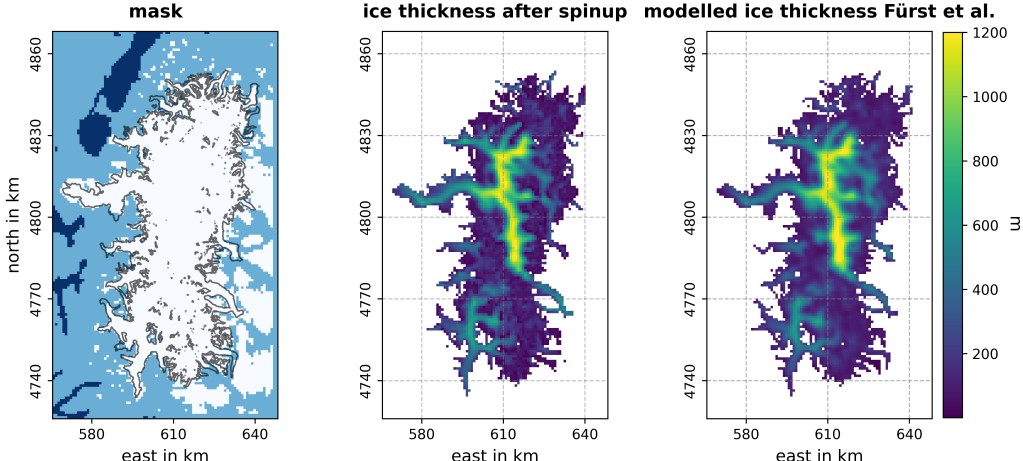

**Figure 4.** Comparison of the NPI state after the spin-up to observations; left: modelled mask (white: ice, light blue: land, dark blue: ocean) and observed year 2000 outlines (black contour), middle: modelled ice thickness after the spin-up clipped to the year 2000 outlines for a better comparison, right: modelled ice thickness for the year 2000 by Fürst et al. (2024).

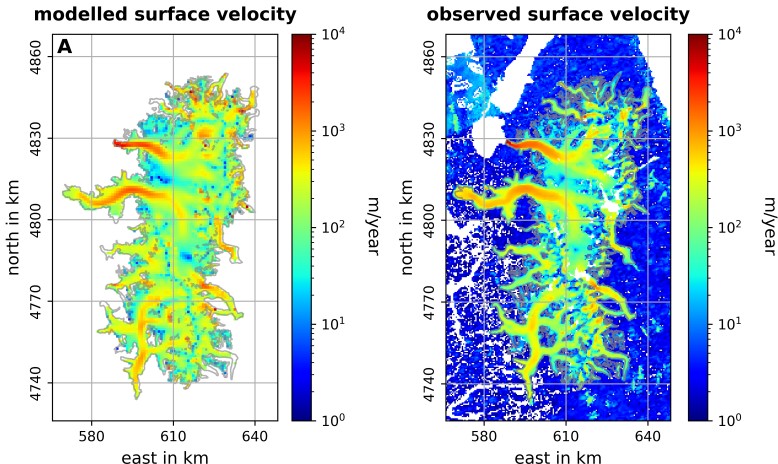

**Figure 5.** Comparison of the NPI surface velocities after the spin-up to observations; left: modelled velocities after the spin-up clipped to the year 2000 outlines; right: observed surface velocities by Mouginot and Rignot (2015), which are centred around the year 2004. Observed year 2000 glacier outlines are shown in grey





## 4 Results

### 4.1 Model calibration

In order to obtain model results that are most similar to the observations in the validation period 2000–2020, we systematically
varied the enhancement factor $E$ of the flow law and the empirical constant $C_b^0$ of the basal sliding coefficient (Eq. (2)). Table 2
summarizes the performance of the different model calibration runs.

| Run name | $E$ | $C_b^0$ $(\mathrm{m\,a^{-1}\,Pa^{-1}})$ | Spin-up volume (% obs. 2000 volume) | Bias speed $(\mathrm{m\,a^{-1}})$ | RMSD speed $(\mathrm{m\,a^{-1}})$ | Bias $\mathrm{d}H/\mathrm{d}t$ $(\mathrm{m\,a^{-1}})$ | RMSD $\mathrm{d}H/\mathrm{d}t$ $(\mathrm{m\,a^{-1}})$ |
|---|---|---|---|---|---|---|---|
| cal1 | 1.4 | $1.8 \times 10^{-4}$ | 103 | 157 | 381 | 0.30 | 2.0 |
| cal2 | 1.4 | $2.0 \times 10^{-4}$ | 101 | 160 | 388 | 0.31 | 2.7 |
| cal3 | 1.4 | $2.2 \times 10^{-4}$ | 100 | 163 | 386 | 0.31 | 2.7 |
| cal4 | 1.5 | $1.8 \times 10^{-4}$ | 101 | 158 | 397 | -0.1 | 4.1 |
| cal5 | 1.5 | $2.0 \times 10^{-4}$ | 100 | 163 | 399 | 0.31 | 2.5 |
| **cal6** | **1.5** | $\mathbf{2.2 \times 10^{-4}}$ | **99** | **165** | **410** | **0.08** | **2.9** |
| cal7 | 1.6 | $1.8 \times 10^{-4}$ | 100 | 167 | 454 | 0.16 | 2.4 |
| cal8 | 1.6 | $2.0 \times 10^{-4}$ | 99 | 168 | 419 | 0.21 | 3.0 |
| cal9 | 1.6 | $2.2 \times 10^{-4}$ | 98 | 167 | 413 | -0.09 | 4.2 |

**Table 2.** Results of the calibration runs with varied enhancement factor $E$ and basal sliding coefficient $C_b^0$. "Bias" and "RMSD" refer to the
linear and quadratic mean, respectively, of the differences between the model results and the observations ("bias" is therefore positive if the
model results are higher than the observations).

We argue that the most important quantity for calibration is the Bias of the elevation change (which is proportional to the
overall ice loss). Therefore, we choose the parameters used in the run `cal6` for the projections. In Figure 6, we visualize the
modelled and observed elevation change using these model parameters.

The prescribed calving rate $a_{\mathrm{calv}}$ (that only affects the front of the San Rafael Glacier; Sect. 3.1) influences the overall
distribution of the ice thickness, thickness change and surface velocity only to a very small extent. The calibration of this
quantity can therefore be done independently of the variations of the enhancement factor and the basal sliding coefficient. A
value of $a_{\mathrm{calv}} = 1000\,\mathrm{m\ ice\ equiv.\ a^{-1}}$ (applied over the entire area of glaciated ocean-facing grid cells) was found to match
the current observed calving flux for the glacier (Minowa et al., 2021) well.

### 4.2 Committed mass change and projections under climate change scenarios

Once having determined the "best parameter set", we realize future simulations starting from the state generated by the spin-up
run (Figures 4, 5). For the committed mass loss run, we use a constant SMB that corresponds to the 1975–2011 mean modelled
one by Schaefer et al. (2013). For the climate change scenarios, we use temperature projections from 33 CMIP6 models (see
Table A1 and Sect. 3.2.2). For comparison, the evolution of the NPI is also projected using SMB projections obtained from




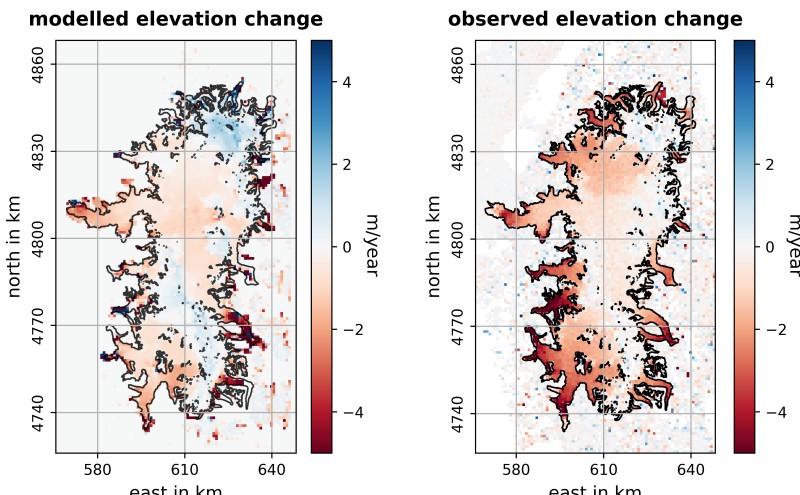

**Figure 6.** NPI modelled (left) and observed (right) elevation changes for 2000-2014.

the simulations of Schaefer et al. (2013) driven by the ECHAM5 model (A1B scenario). The results of the projection are summarized in Figure 7.

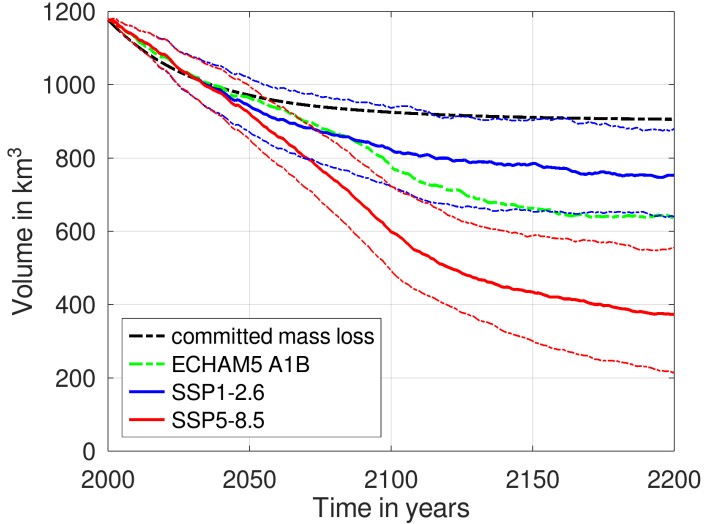

**Figure 7.** Simulated evolution of the total ice volume of the NPI during the 21st and 22nd centuries under a constant current climate (black dashed), the SSP1-2.6 scenario (blue lines) and the SSP5-8.5 scenario (red lines). The continuous lines are based on global circulation model mean temperature anomalies, and the dashed lines on mean ± one standard deviation. For comparison, the projection using SMB anomalies generated with ECHAM5 data under the old A1B scenario is also shown (green dashed).




The committed mass loss run projects approximately one-quarter of the initial (year 2000) mass to be lost by 2200, while the loss is more than one-third for SSP1-2.6 and more than two-thirds for SSP5-8.5. In Figures 8 and 9  we show the evolution

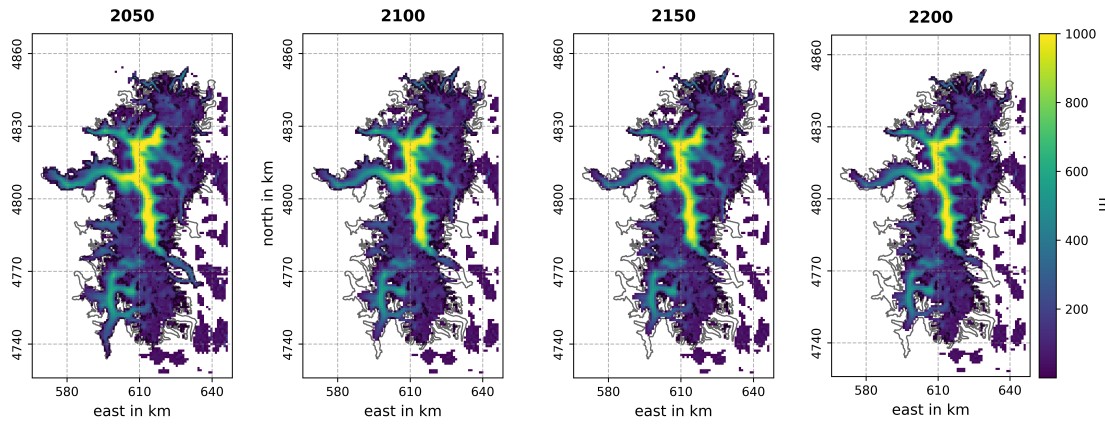

**Figure 8.** Simulated ice thickness of the NPI in the 21st and 22nd centuries under the SSP1-2.6 scenario.

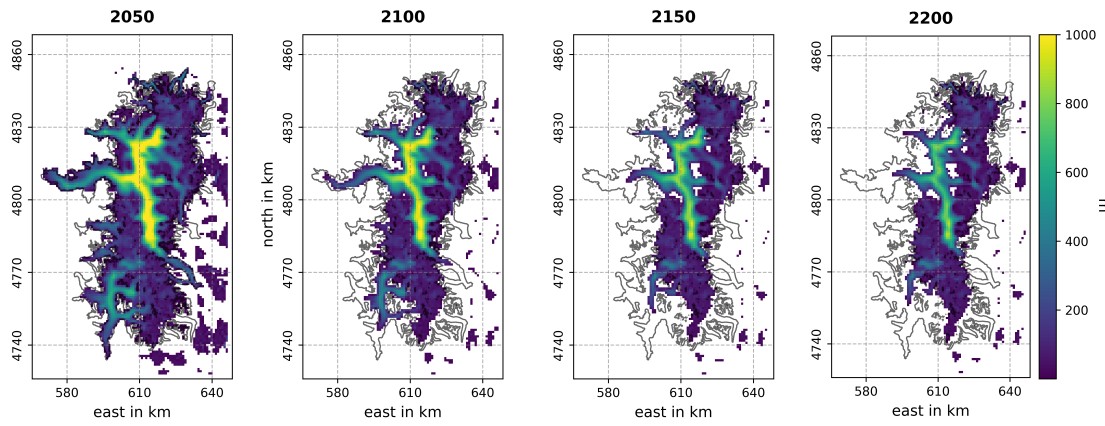

**Figure 9.** Simulated ice thickness of the NPI in the 21st and 22nd centuries under the SSP5-8.5 scenario.

of the ice thickness distribution under the climate change scenarios SSP1-2.6 and SSP5-8.5. For SSP1-2.6, all glaciers retreat, and several glacier tongues disappear by the end of the 22nd century. However, the ice thickness remains similar in the interior part of the NPI, which contains the great plateau between San Rafael Glacier and Colonia Glacier (Figure 8). By contrast, for SSP5-8.5, the extent of the icefield, as well as the ice thickness in the interior, strongly reduce until 2200 (Figure 9).



## 5 Discussion

Our results provide the first projections for the evolution of the NPI as a whole using a three-dimensional ice-flow model.
Important differences in comparison to previous projections for or on the NPI are:

- the implementation of a calving law, which lets the ice front freely evolve,

- the simulations were run until 2200, which lets the icefield adjust to the strong expected climate trends during the 21st century,

- the use of SMB data that are consistent with direct observations.

In Figure 10, we compare our ice volume projections to other studies. The only study that provides projections for the NPI is

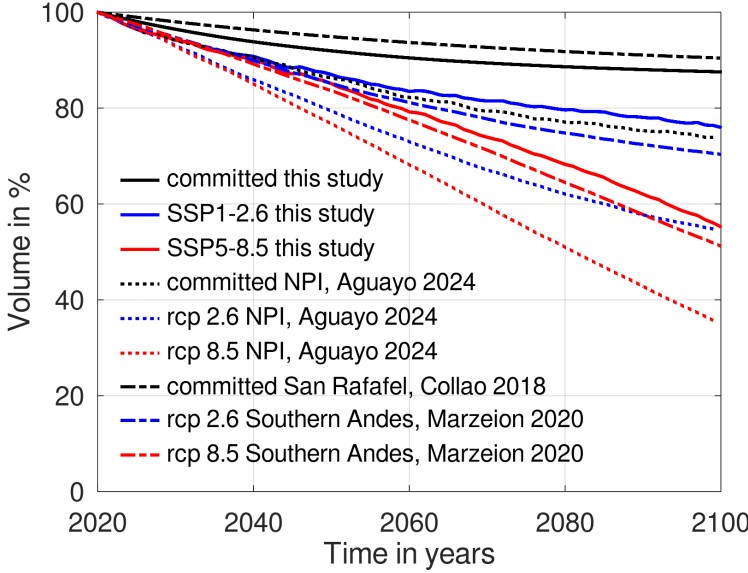

**Figure 10.** Projection of relative glacier volume change in the Southern Andes throughout the 21st century, relative to 2020, from different studies.


Aguayo et al. (2024), who used the Open Global Glacier Model (OGGM) (Maussion et al., 2019). They project much higher mass loss during the 21st century than our study, and also compared to the projections for the Southern Andes by the Glacier Model Intercomparison Project (Marzeion et al., 2020). We think the reason for this is that calving is not implemented in the OGGM version used by Aguayo et al. (2024). Since the model is calibrated against geodetic mass balances (which include

calving losses), the temperature sensitivity parameter necessary to reproduce the negative mass balance of the NPI during their calibration period (2000–2020) is probably chosen too high, which explains the high mass losses projected for the 21st century. Although Aguayo et al. (2024) probably present reliable projections for glacier regions in Patagonia with smaller





glaciers, which do not experience important mass losses by calving, their results for the large icefields of Patagonia (NPI, SPI, Cordillera Darwin), where frontal ablation contributes up to 48% to total ablation (Minowa et al., 2019), should be interpreted

with much care.

The good agreement of the relative volume evolution projections of our study with the results of the Glacier Model Inter-comparison Experiment (Marzeion et al., 2020) is interesting, but should also be interpreted with some care. Although the NPI is one of the most important ice bodies in the Southern Andes, the SPI holds approximately 3.5 times the ice volume of the NPI (Fürst et al., 2024). The projection of this huge ice body should dominate the projections of the ice volume of the Southern

Andes. The more moderate projected mass losses compared to Aguayo et al. (2024) are probably due to the implementation of calving losses in some of the global glacier models (Huss and Hock, 2015) and the use of direct SMB observations (instead of geodetic) for model calibration.

The evolution of the relative ice volume of the entire NPI in our committed mass loss run is similar to the results obtained by Collao-Barrios et al. (2018) for the San Rafael Glacier. This is somehow surprising, regarding that Collao-Barrios et al.

(2018) used fixed glacier outlines and an SMB that is much lower than the one used in our study. One explanation is that, in the committed mass loss run, similar to the SSP1-2.6 scenario, the modelled retreat of San Rafael Glacier is very moderate. On the other hand, Collao-Barrios et al. (2018) tuned the SMB to observed elevation changes in their calibration period (2000–2012), which is similar to the time span of the observed elevation changes used in our study (2000–2014). Of course, using SMB parameters for model tuning can produce reasonable results for the committed mass loss under constant climate, but makes it

very difficult to realize projections under changing climate.

Comparing our results to the projections of Scheiter et al. (2021) for the Mocho-Choshuenco ice cap, also located in the Wet Andes, their projected relative ice volume losses by the end of the century of $56 \pm 16\%$ for RCP2.6 and $97 \pm 2\%$ for RCP8.5 are much higher. This is in line with the results of global studies, where glacier regions with smaller glaciations experience higher percentage losses (see, e.g., Marzeion et al. (2020)). As a consequences of this the relative ice volume losses should be

significantly higher for the Dry Andes as for the Wet Andes. However, this distinction has not been captured well by global studies to date, as they typically group the highly contrasting Dry Andes and Wet Andes regions into a single category referred to as the "Southern Andes" (RGI Consortium, 2017).

Another interesting result of our study is that important glacier changes will continue during the 22nd century even under a constant late-21st-century climate (no further warming trend beyond 2100). We can compare this result to the theoretical

formula for the glacier response time $t_r$ (Cuffey and Paterson, 2010),

$$t_r = \frac{H_{\max}}{a_0}, \tag{3}$$

where $H_{\max}$ is the maximum ice thickness and $a_0$ the ablation rate at the glacier tongue. From our input data, we find $\bar{a}_0 = 15.6 \,\mathrm{m\ ice\ equiv.\ a^{-1}}$, where we took an average over modelled ablation values at the tongues of the 17 most important glaciers of the NPI (Schaefer et al., 2013), and $H_{\max} = 1440 \,\mathrm{m}$ (Fürst et al., 2024), which results in $t_r = 92.3 \,\mathrm{a}$. This value is in good

agreement with the continuing changes observed in our simulation under a constant climate during the 22nd century. A similar values was found by (Zekollari et al., 2025) for the Southern Andes Glacier region. This result indicates that the interpretation

of observed mass changes (and their direct association with changes in climate, Noël et al. (2025)) should be realized with caution since the observed mass changes may be influenced by climate changes that occurred in a time span of nearly 100 years.

This result also leads us to the main caveat of our study. The validation period is shorter than the expected response time. In future studies, longer calibration periods should be used to improve confidence in the results. Simulations since the Little Ice Age for which different glacier outline products are available (Glasser et al., 2011; Meier et al., 2018) are desirable. The problem of such simulations is the large uncertainty of the initial glacier state with respect to ice elevations and surface velocities. As a main contributor to ice volume changes in the Wet Andes (and the Southern Andes), a similar modelling effort

should be realized for the SPI. Due to the much higher relative mass loss by frontal ablation there (48%, Minowa et al. (2021)), probably more sophisticated parameterizations of calving losses will have to be used in the ice-flow model.

## 6   Conclusions

In this contribution, we realize projections for the NPI under different climate change scenarios using a state-of-the-art ice-flow model including a simple implementation of frontal ablation for the tidewater glacier San Rafael. Our most important

conclusions are as follows:

- large volume changes are projected until the year 2200 with important differences depending on the climate change scenario (36% of ice loss in SSP1-2.6 and 68% in SSP1-8.5),

- the NPI has a response time of approximately 100 years, which should be taken into account when interpreting current changes,

- future studies should consider larger time spans for model calibration and focus on the largest ice body of the Southern Andes, the Southern Patagonia Icefield.

## Appendix A: Supplementary table



**Table A1.** Selected CMIP6 models and their projected 2090–2099 mean temperature anomaly (°C) with respect to years 2000–2015 for scenarios SSP1-2.6 and SSP1-8.5 and averaged over the NPI.

| GCM | SSP1-2.6 | SSP5-8.5 |
|---|---|---|
| ACCESS-CM2 | 1.23 | 3.46 |
| AWI-CM-1-1-MR | 0.32 | 2.92 |
| BCC-CSM2-MR | 0.77 | 1.94 |
| CAMS-CSM1-0 | 0.57 | 1.58 |
| CanESM5 | 0.81 | 3.60 |
| CanESM5-CanOE | 0.62 | 3.56 |
| CESM2 | 1.10 | 3.73 |
| CIESM | 1.57 | 4.47 |
| CMCC-CM2-SR5 | 0.54 | 2.47 |
| CMCC-ESM2 | 0.85 | 2.54 |
| CNRM-CM6-1 | 1.35 | 3.69 |
| CNRM-CM6-1-HR | 1.60 | 4.16 |
| CNRM-ESM2-1 | 0.83 | 3.16 |
| FGOALS-f3-L | 1.00 | 2.98 |
| FGOALS-g3 | 0.14 | 1.73 |
| FIO-ESM-2-0 | 0.92 | 3.78 |
| GFDL-ESM4 | 0.21 | 2.01 |
| HadGEM3-GC31-LL | 1.08 | 3.83 |
| HadGEM3-GC31-MM | 2.00 | 4.66 |
| IITM-EM | 0.83 | 2.17 |
| INM-CM4-8 | 0.33 | 1.89 |
| INM-CM5-0 | 0.26 | 1.75 |
| IPSL-CM6A-LR | 0.85 | 3.33 |
| KACE-1-0-G | 1.27 | 3.31 |
| KIOST-ESM | -0.04 | 1.71 |
| MIROC-ES2L | 0.23 | 1.77 |
| MIROC6 | 0.19 | 1.92 |
| MPI-ESM1-2-LR | 0.51 | 1.97 |
| MRI-ESM2-0 | 0.73 | 2.76 |
| NESM3 | 0.21 | 2.18 |
| NorESM2-MM | 0.71 | 1.84 |
| TaiESM1 | 1.61 | 4.13 |
| UKESM1-0-LL | 0.85 | 3.73 |
| Mean | 0.79 | 2.87 |



*Code and data availability.* SICOPOLIS (SICOPOLIS Authors, 2025) is free and open-source software, published on a persistent Git repository hosted by GitHub (https://github.com/sicopolis/sicopolis/). Detailed instructions for obtaining and compiling the code are at

https://www.sicopolis.net (last access: 5 June 2025). The output data produced for this study will be made available at Zenodo on publication.

*Author contributions.* M.S. designed the study with input from all coauthors. I.T. extracted the CMIP6 temperature projections over the NPI and implemented the calving parameterization in SICOPOLIS. M.S. and R.G. conducted the SICOPOLIS simulations. All authors contributed to the discussion and interpretation of the results. The manuscript was written by M.S. with contributions from all coauthors.

*Competing interests.* J.F. is a member of the editorial board of The Cryosphere

*Acknowledgements.* We thank David Farias and Christian Sommer for sharing elevation change data of the NPI. We also thank the support team of the Erlangen National High Performance Computing Center.

*Financial support.* Marius Schaefer's stay at the Friedrich-Alexander University of Erlangen–Nuremberg was financed by the Humboldt Research Fellowship for Experienced Researchers (CHL 1216763 HFST-E). Ilaria Tabone was supported by the DFG Individual

Research Grant Number 495516510. Ralf Greve was supported by Japan Society for the Promotion of Science (JSPS) KAKENHI Grant Nos. JP17H06104 and JP17H06323. Johannes Fürst was supported by the FRAGILE project (ERC-2020 StG 948290). Matthias Braun and Johannes Fürst are supported by the German Research Foundation (DFG) project ITERATE (BR 2105/28-1 & FU 1032/12-1).



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
