# Peer review of "Projecting the evolution of the Northern Patagonian Icefield until the year 2200"

_EGUsphere, 2025_

## Referee Comment (RC2)

**Review of "Projecting the evolution of the Northern Patagonian Icefield until the year 2200" by Schaefer et al.**

This manuscript presents simulations of the Northern Patagonian Icefield and its future evolution through the year 2200. The manuscript is generally well written, clearly structured, and suitable for publication in *The Cryosphere*. However, I found that the Results and Discussion sections could be strengthened, particularly regarding the treatment and discussion of uncertainties. My main comments are outlined below.

| Genera | lc | ٥m  | m | ۵n | te: |
|--------|----|-----|---|----|-----|
| Genera |    | UII |   |    | LJ. |

\_\_\_\_\_\_

**SMB**

If I understand correctly, you apply a present-day surface mass balance (SMB) from Schaefer et al. (2013) and adjust it homogeneously using a temperature anomaly. If so, precipitation changes derived from the Earth System Models (ESMs) are not accounted for. I would expect that under warmer conditions, increased melt would occur but potentially also increased precipitation at higher elevations. Is it not feasible to run the reference experiments using precipitation fields directly from the ESMs you employ?

Related to this, I suggest addressing the following points in the Discussion section:

- Do you expect the SMB forcing approach used here to respond similarly across the range of ESMs considered?
- Are elevation feedbacks included in your SMB-temperature relationship? From the current description, it does not appear that they are.
- Spatial patterns present in the ESMs may not be captured with the adopted methodology. I understand that downscaling ESM outputs may be challenging, but acknowledging this limitation would strengthen the discussion.
- What is the temperature spread of each ESM over the Northern Patagonian Icefield? As I understand it, the standard deviation you apply corresponds to the variability of the mean temperature values across models. However, I would expect substantial spatial variability in temperature within the domain. In that case, applying a spatially uniform standard deviation may not fully capture the uncertainty. Would it be more appropriate to use a spatially varying measure of variability instead?

**Spin-up**

I was also uncertain about certain aspects of the spin-up procedure. As I understand it, you use the SMB field from Schaefer et al. (2013) but increase the input SMB by reducing global temperature by 1°C—is this correct? Additionally, it seems that only one spin-up is performed during 500 years, and then followed by 20-year runs with varying parameters for

calibration. In that case, would part of the diagnosed drift between 2000–2020 arise from parameter changes rather than the applied forcing? Some clarification would be useful.

**Basal friction**

Regarding the basal friction law, you define a relationship dependent on basal temperature and basal water thickness, and Figure 2 shows the effect of varying these quantities. However, no sensitivity experiments are presented for these parameters. The spin-up varies only  $C_b^0$ , while  $C_w$ ,  $\gamma$ , and  $H_w$  are fixed. Under this configuration, the relevance of Figure 2 is unclear, and removing it may improve the manuscript's focus.

That said, I believe a sensitivity analysis of H\_w or geothermal heat flux would be more informative than focusing solely on friction coefficients, because these could significantly influence basal sliding. Could you include at least one or two sensitivity tests to illustrate this?

**Parameter values**

I also recommend citing references supporting your chosen values for C\_w, γ, and H\_w. Similarly, the source of the geothermal heat-flux value should be given. Different choices for geothermal heat flux could warm or cool the bed and potentially alter the dynamical state, so discussing this in the manuscript would be valuable.

**Calving law**

Concerning calving: How is the calving front represented in the model? Do you employ a level-set method, or do you apply a basal melting rate to ice-front nodes? The applied calving rate of 1000 m/yr appears high. When you say this value "was found to match the current observed calving flux," do you refer to flux magnitude or to the present-day terminus position? If it refers to flux, please provide the corresponding observed calving flux value.

**Future scenarios**

It appears that only one calibration ensemble member (cal6) is used for the forcing. Relying on a single member may limit the robustness of the conclusions. Since all ensemble members show broadly similar present-day tendencies, could additional members be included to assess the sensitivity of future projections? You could add a weight to these simulations based on their present-day performance. This may also help evaluate the influence of parameters such as friction coefficients or enhancement factors.

**Comments on Figures**

• **Figures 4 and 5:** Instead of plotting model results alongside observations, it may be more informative to show the anomaly (model – observation). This could highlight spatial biases that are otherwise difficult to identify.

- **Figure 7:** Consider including the simulated ice-covered area for completeness.
- **Figure 10:** It appears similar to Figure 7 but expressed in percentages. If so, you may consider merging or simplifying.

**Minor Comments**

- **Table 1:** Please clarify the role of the "residual stress parameter." I could not find further explanation in the manuscript.
- Line 272: It should be SSP5-8.5, not SSP1-8.5 (also in Table A1).
- I agree with the other reviewer that removing the ECHAM5 scenario would improve clarity and facilitate the reading of the manuscript.

---

## Author Comment (AC1)

Review of "Projecting the evolution of the Northern Patagonian Icefield until the year 2200" by Schaefer et al.

This manuscript presents simulations of the Northern Patagonian Icefield and its future evolution through the year 2200. The manuscript is generally well written, clearly structured, and suitable for publication in The Cryosphere. However, I found that the Results and Discussion sections could be strengthened, particularly regarding the treatment and discussion of uncertainties. My main comments are outlined below.

General comments:
=======================================================================
SMB

If I understand correctly, you apply a present-day surface mass balance (SMB) from Schaefer et al. (2013) and adjust it homogeneously using a temperature anomaly. If so, precipitation changes derived from the Earth System Models (ESMs) are not accounted for. I would expect that under warmer conditions, increased melt would occur but potentially also increased precipitation at higher elevations. Is it not feasible to run the reference experiments using precipitation fields directly from the ESMs you employ?

**The focus of our work is the application of a ice-flow model to the NPI. Regarding the surface mass balance this work strongly relies on Schaefer et al. 2013. In Schaefer et al. a very clear relationship between projected (mean) surface temperature over the NPI and the (mean) surface mass balance was found ( explained variance of 80%, Figure 3 left panel of our study). In this work we rely on this relationship between temperature and surface mass balance. Using precipitation fields directly from the ESMs is not possible due their low spatial resolution which does not represent at all the complex topography of the NPI (see Figure 1 below). Another good argument for our procedure is that temperature projection are generally more reliable then projection of precipitation.**

Related to this, I suggest addressing the following points in the Discussion section:
● Do you expect the SMB forcing approach used here to respond similarly across the range of ESMs considered?

**We do not apply our smb parametrization to the to temperature projections of the individual ESM but to the mean value of the temperature (anomalies) and to the mean +/- one standard deviation.**

● Are elevation feedbacks included in your SMB–temperature relationship? From the current description, it does not appear that they are.

**Thanks for your question. SICOPOLIS has this feature but we did not not activate it so far. We will activate it for next round of simulations.**

● Spatial patterns present in the ESMs may not be captured with the adopted methodology. I understand that downscaling ESM outputs may be challenging, but acknowledging this limitation would strengthen the discussion.

**The spatial patters of the surface mass balance fields stem from Schaefer et al. 2013 , where model parameters were adjusted to represent best the spatial variation of point surface mass balance measurements form the Eastern and Western side of the icefield including two ice-cores from the accumulation area and geodetic mass balances of three not calving glaciers. Since the spatial resolution of ESMs is very low, a maximum of four grid cells was used to infer future temperature (mean) anomalies (see figure below) We will give detailed information on model resolution in table A1 in the revised version of our manuscript.**

[Figure]

*Figure 1: Illustration of the ESM model grid cells and NPI extension for two ESMs with different spatial resolution.*

● What is the temperature spread of each ESM over the Northern Patagonian Icefield?
**We did not check that explicitly, but as there only two grid cells involved it should be very low!**
As I understand it, the standard deviation you apply corresponds to the variability of the mean temperature values across models. However, I would expect substantial spatial variability in temperature within the domain. In that case, applying a spatially uniform standard deviation may not fully capture the uncertainty. Would it be more appropriate to use a spatially varying measure of variability instead?
**We agree that assuming spatial uniform temperature spread is a simplification of the reality. However as there are only two grids involved we argue that the impact of this simplification should be low.**
Spin-up
I was also uncertain about certain aspects of the spin-up procedure. As I understand it, you use the SMB field from Schaefer et al. (2013) but increase the input SMB by reducing global temperature by 1°C—is this correct?
**Yes!**
Additionally, it seems that only one spin-up is performed during 500 years, and then followed by 20-year runs with varying parameters for calibration. In that case, would part of the diagnosed drift between 2000–2020 arise from parameter changes rather than the applied forcing? Some clarification would be useful.

**For every calibration run a separate spin-up was realized with the same parameter set, so no model drift is expected which stems from parameter changes. We will state this more explicitly in the new version of the manuscript.**

Basal friction

Regarding the basal friction law, you define a relationship dependent on basal temperature and basal water thickness, and Figure 2 shows the effect of varying these quantities. However, no sensitivity experiments are presented for these parameters. The spin-up varies only $C_b^0$, while $C_w$, $\gamma$, and $H_w$ are fixed. Under this configuration, the relevance of Figure 2 is unclear, and removing it may improve the manuscript's focus.

**We think that Figure 2 represents a useful visualization of equation (2), but we also agree that slip parametrizations are not the main focus of our study. We will move the figure to the appendix.**

That said, I believe a sensitivity analysis of $H_w$ or geothermal heat flux would be more informative than focusing solely on friction coefficients, because these could significantly influence basal sliding. Could you include at least one or two sensitivity tests to illustrate this?

**Thank you for your suggestion. We will realize a sensitivity analysis of the parameters you mentioned and inform the results in the new version of our manuscript.**

Parameter values

I also recommend citing references supporting your chosen values for $C_w$, $\gamma$, and $H_w$.

**These parameters were optimized in order to obtain the best agreement between modeled icefield and current (year 2000) NPI state.**

Similarly, the source of the geothermal heat-flux value should be given.

**We chose 65 mW/m² since it is in good agreement with the mean heat flow in South America of 63+-36 mW/m² (Hamza et al. 1996). Reference will be added in the new version of the manuscript.**

Different choices for geothermal heat flux could warm or cool the bed and potentially alter the dynamical state, so discussing this in the manuscript would be valuable.

**Ok, discussion will be added depending on the results of the sensitivity tests.**

Calving law

Concerning calving: How is the calving front represented in the model? Do you employ a level-set method, or do you apply a basal melting rate to ice-front nodes? The applied calving rate of 1000 m/yr appears high. When you say this value "was found to match the current observed calving flux," do you refer to flux magnitude or to the present-day terminus position? If it refers to flux, please provide the corresponding observed calving flux value.

**Inferred calving fluxes in literature range from 0.6 to 1.85 Gt/year. Minowa et al 2021 found 1 Gt/year. With our our current calving parameter a_calv=1000m/year at 900m grid cells size the maximum model calving flux per grid cell is 0.9x0.9x1.0Gt=0.81 Gt/year. During the calibration ( as well in the projection) period the ice tongue is mostly represented by two grid cells which would give a maximum calving flux of 1.62 Gt/year. The "real" calving fluxes are often lower then this value since sometimes there is not enough ice to calve of ( no negative ice thickness is allowed).   More details on the calving parametrization and model resolution will be given in the re-submission of our manuscript.**

Future scenarios

It appears that only one calibration ensemble member (cal6) is used for the forcing. Relying on a single member may limit the robustness of the conclusions. Since all ensemble members show broadly similar present-day tendencies, could additional members be included to assess the sensitivity of future projections? You could add a weight to these

simulations based on their present-day performance. This may also help evaluate the influence of parameters such as friction coefficients or enhancement factors.

**Thank you for this suggestion. We will test how the projection changes when using other calibration parameter sets ( e.g. cal4 and cal9, which have bias in dH/dt with a similar magnitude). Depending on the result of this sensitivity test we will decide if and how to incorporate different calibration parameter sets in our projections.**

Comments on Figures
● Figures 4 and 5: Instead of plotting model results alongside observations, it may be more informative to show the anomaly (model − observation). This could highlight spatial biases that are otherwise difficult to identify.
**Thank you for the suggestion. We will show the differences in the new version of our manuscript.**
● Figure 7: Consider including the simulated ice-covered area for completeness.
**We will add information on area changes in a new table in the supplementary part of the paper and add an area evolution plot in the supplementary as well.**
● Figure 10: It appears similar to Figure 7 but expressed in percentages. If so, you may consider merging or simplifying.
**Yes, both figures look similar but fulfill a very different purpose:   Figure 7 presents the results of our simulations, but figure 10 compares our results to the results of other contributions. Relative volume values are shown here since we are also comparing to the simulation with include other icebodies.**
Minor Comments
● Table 1: Please clarify the role of the "residual stress parameter." I could not find further explanation in the manuscript.
**The residual stress parameter appears in the regularized Glen flow law, which avoids the infinite-viscosity limit for zero effective stress (Greve and Blatter 2009, Sect. 4.3.2).**
● Line 272: It should be SSP5-8.5, not SSP1-8.5 (also in Table A1).
**Ok thanks, changed!**
● I agree with the other reviewer that removing the ECHAM5 scenario would improve clarity and facilitate the reading of the manuscript.
**Ok, we will remove the Echam5 model in Figure 3, right panel and in Figure 7.**

**References:**

**Greve, R. and Blatter, H.: Dynamics of Ice Sheets and Glaciers, Springer, Berlin, Germany etc., https://doi.org/10.1007/978-3-642-03415-2, 340, 2009**

**Hamza, V. M. and Muñoz, M.: Heat flow map of South America, Geothermics, 25, 599–646, https://doi.org/https://doi.org/10.1016/S0375-6505(96)00025-9, 1996**

---

## Author Comment (AC2)

Review of Schaefer et al, The Cryosphere

Thank you for this opportunity to review this manuscript, and apologies for the tardy review.

**General comments**

Schaefer et al present a most welcome analysis of the future of the Northern Patagonian Icefield, a sensitive icefield of some import in Patagonia. The manuscript is clearly presented and provides a careful analysis of the future behaviour of the icefield.

Thank you!

I have some recommendations for improved clarity and to hopefully help elevate the manuscript.

My most significant comment is to do with the surface mass balance approximation of the icefield. The Andes have one of the most extreme climatic divides found worldwide (Sauter, 2020). The strong orographic influence on climate is evident in terrestrial observations. To what extent does downscaled climate data match observations? This needs a clear evaluation.

The main focus of this contribution is the application of an ice flow model to the Northern Patagonia Icefield and realize projections. Regarding climate data and corresponding surface mass balance maps this work relies strongly on the results of Schaefer et al 2013. Comparison between modeled and observed climate are shown in Table 2 and Figures 5, 6, and 7 of this contribution. Temperature biases of the modeled data range from -0.13 and +1.14 deg. Celsius when comparing to individual weather stations. Correlation of monthly precipitation time series range from 0.21 (in El Calafate) to 0.81 (Puerto Aysen). The climate divide is nicely visible in the modeled incoming radiation data (Figure 8 of Schaefer et al. 2013), where the glacier tongues on the Eastern side receive considerably more radiation then the plateau area of NPI.

**Specific comments**

Bed topography – how does the bed topography compare with that provided by Millan et al. (2022)?

A comparison of both datasets is shown in Fuerst et al 2024 (Fig 3). The bed we used has deeper and more connected troughs then the one obtained from Millan et al.  We could add a comparative plot in the supplementary.

In line 136 insufficient detail is provided for the SMB model. How does this calculate SMB? This is important, as differences in SMB may explain the differences in these projections and others in the Discussion.

More details on the surface mass balance model were given now.

Line 138 – the ablation stakes and shallow firn cores are vague, and this evaluation needs more explanation.

More details on the ablation stakes and firn cores and the model validation will be indicated in the new version of our manuscript..

Line 153 – how do these ice velocity measurements differ to Millan et al. (2022)?

**The study period of Mouginot and Rignot (2018) is 1984-2014 whilst Millan et al. (2022) are studying 2017-2018.**

The modelling workflow is provided clearly, but some more details are required in places.

Line 169 – I agree that the NPI in 2000 was likely not in equilibrium with climate, and there observational datasets of ice recession and thinning at this time that would support this statement.

**Data on ice recession and thinning are used in the model calibration phase. The results of this phase are presented in section 4.1: Table 2 & Figure 6**

Line 197-198 – to help the reader, clarify again the experiments run (SSPs, to what year, etc.).

**Ok, sentences will be rephrased and information on SSPs will be added.**

Line 200 – it is not clear to me why the additional experiment forced by ECHAM5 A1B scenario is used. This is quite dated now and has been superseded by CMIP6. Also this is one model, whereas the CMIP6 is a multi model mean. I would recommend just removing this experiment as I don't think it adds anything.

**Ok, the outdated scenario was shown in order make the link to the results of Schaefer et al. 2013. It will be removed in the revised version of the manuscript.**

Line 202 – to help the reader, and improve usefulness for readers, I recommend also summarising the volume and area change as a % of the 2000 ice volume (or some other standard year).

**Volume changes until 2100 as a % of the 2020 ice volume are shown in Figure 10. Two new tables will be added in the supplementary where % of the 2000 ice volume and icefield area values are indicated for the years 2050, 2100, 2150 and 2200 for both SSPs and the committed mass loss run.**

Figure 8/9 – the 'm' in the colour bar is rotated. It would be easier to read if it were horizontal. Note the year of glacier outlines shown (2000?).

**The unit of the colorbars is rotated in all maplots (Figures 4,5,6,8,9) . This mostly for space efficiency (when the unit is m/year). For consistency we would prefer to keep it rotated. Yes the outlines correspond to the year 2000 outline, information will be added.**

Clarify here why the experiments were run to 2200 and not 2300, when CMIP 6 extended model runs are available until 2300 CE (Eyring et al., 2016).

**We think that the climate of the 22nd and 23rd is very uncertain and depends crucially on the decisions the humanities takes in the ongoing century. We prefer to use instead a constant climate forcing from 2100 on. Simulation were stopped in 2200 as the ice volume got mostly stable in the simulation by 2200.**

Line 204 – clarify here the total area loss in km2 and % change

**The area losses will summarized in Table A3 and and reference to the table will be added in the text.**

Overall I found the results section rather brief. I would add more details on glacier changes under the two different scenarios. Do glaciers remain calving by 2200 or are all on dry land, above sea level? What is the ice velocity?

**Thank you for asking these interesting questions. In the SSP1-2.6 scenario San Rafael Glacier will remain a tidewater calving glacier until the end of the simulation period, whilst in SSP5-8.5 scenario it turns into a land-terminating glacier. Ice thickness reduction is accompanied by ice speed reduction. Information on both points will be added in the results section.**

Can you show the simulated surface mass balance on the icefields for the two different SSPs at different timescales (like the ice thickness figure).

**The surface mass balance which corresponds to the ice-thickness states shown in Figure 8 and 9 will be shown in two additional figures in the supplementary.**

Line 217 – more information on the SMB model used could also be helpful here. Was the SMB a temperature index model or something more complex, and could this explain the differences?

**Aguayo et al. 2024 use a temperature index model, which could also be a reason for the observed differences. This will be mentioned in text.**

Line 221 – precipitation is crucial for the Patagonian icefield SMB (Sauter, 2020); any biases in the downscaled climate data would have a big effect, potentially larger than calving flux. This needs more careful evaluation.

**We agree that solid precipitation input is crucial for the temperate Patagonian Glacier. However there is very few information on the amounts of precipitation falling over the Patagonian Icefields. Schaefer et al. 2013 compared their SMB simulation to firn cores taken on the NPI and obtained satisfactory results (Figure 11).**

Line 227 – could these differences also be explained because GlacierMIP2 (Marzeion et al., 2020) is forced by CMIP6 at a broader scale, with temperature index models used to calculate SMB at best. How much more (or less) reliable is the climate forcing data used here given the statistical downscaling?

**In this section we say rather try to say that our results are "surprisingly similar" to GlacierMIP2 and indicate that this is rather a comparison then a validation. The comparison is a bit comparing apples with oranges, but we still think that has some value, since we do not have more apples in the region ...**

Line 272 – I couldn't see these results of % change in the results, don't include new things in the conclusions and put these data in the results section too.

**The % changes will be added in Table A2 and mentioned in the results section.**

Table A1 – precipitation variation will also be very important, can you also provide this information in the table?

**Our surface mass balance projections are solely based on temperature anomalies ( see section 3.2.2 and Figure 3, left panel), so we did not analyze precipitation variation.**

**References used in this review.**

Eyring, V., Bony, S., Meehl, G.A., Senior, C.A., Stevens, B., Stouffer, R.J., Taylor, K.E., 2016. Overview of the Coupled Model Intercomparison Project Phase 6 (CMIP6) experimental design and organization. *Geosci. Model Dev.* 9, 1937-1958.

Marzeion, B., Hock, R., Anderson, B., Bliss, A., Champollion, N., Fujita, K., Huss, M., Immerzeel, W., Kraaijenbrink, P., Malles, J.-H., Maussion, F., Radić, V., Rounce, D.R., Sakai, A., Shannon, S., van de Wal, R., Zekollari, H., 2020. Partitioning the Uncertainty of Ensemble Projections of Global Glacier Mass Change. *Earth's Future* 8.

Millan, R., Mouginot, J., Rabatel, A., Morlighem, M., 2022. Ice velocity and thickness of the world's glaciers. *Nature Geoscience* 15, 124-129.

Sauter, T., 2020. Revisiting extreme precipitation amounts over southern South America and implications for the Patagonian Icefields. HH*ydrol. Earth Syst. Sci.* 24, 2003-2016.